# Stimulus-responsive assembly of nonviral nucleocapsids

Mao Hori[1,2,5], Angela Steinauer ⓘ[1,3,5], Stephan Tetter ⓘ[1,4], Jamiro Hälg ⓘ[1], Eva-Maria Manz ⓘ[1] & Donald Hilvert[1] ✉

Controlled assembly of a protein shell around a viral genome is a key step in the life cycle of many viruses. Here we report a strategy for regulating the co-assembly of nonviral proteins and nucleic acids into highly ordered nucleo-capsids in vitro. By fusing maltose binding protein to the subunits of NC-4, an engineered protein cage that encapsulates its own encoding mRNA, we successfully blocked spontaneous capsid assembly, allowing isolation of the individual monomers in soluble form. To initiate RNA-templated nucleocapsid formation, the steric block can be simply removed by selective proteolysis. Analyses by transmission and cryo-electron microscopy confirmed that the resulting assemblies are structurally identical to their RNA-containing counterparts produced in vivo. Enzymatically triggered cage formation broadens the range of RNA molecules that can be encapsulated by NC-4, provides unique opportunities to study the co-assembly of capsid and cargo, and could be useful for studying other nonviral and viral assemblies.

Cooperative self-assembly is a fundamental principle underlying diverse biological phenomena. A striking example is the construction of viruses, which comprise many copies of a single protein that self-assemble to package viral genomes during replication[1]. Taking a cue from nature, molecular engineers have harnessed the cooperative self-assembly of biological polymers to create a wide range of novel DNA, RNA, and protein-based nanostructures[2–4]. Because of their well-defined structures and amenability to genetic and chemical modification, protein cage architectures, including natural virus capsids, ferri-tins, and bacterial microcompartments, have played a prominent role in such efforts[5–7]. The properties of these molecular containers can be readily tailored using the tools of protein engineering. Advances in computational protein design have also enabled the bottom-up con-struction of many non-natural supramolecular assemblies[8–10]. Like their natural counterparts, engineered protein cages have proven useful as molecular containers for diverse cargo molecules, including proteins, nucleic acids, metal nanoparticles, quantum dots, and low molecular weight drugs[11–13]. Not surprisingly, practical applications in drug deliv-ery, imaging, therapy, and other areas are being actively explored[13–16].

The capsid-forming enzyme lumazine synthase from the hyper-thermophilic bacterium *Aquifex aeolicus* has been extensively explored in this context[17]. This 60-subunit, icosahedrally symmetric nanocompartment catalyzes the penultimate step in the biosynthesis of the vitamin riboflavin and increases overall metabolic efficiency by encapsulating the next enzyme in the pathway[18]. This scaffold is exceptionally malleable, however, and has been diversified through design and evolution to afford a family of unique proteinaceous cap-sules of different sizes, symmetries, and morphologies[19–22]. These modified cages have also been assembled with other components to produce delivery vehicles, imaging agents, nanoreactors, artificial organelles, and virus mimics[23–33]. Providing spatiotemporal control over the assembly and disassembly of such structures could provide an additional means of regulating both their structure and functionality.

In this work, inspired by recent efforts to develop natural and artificial protein cages whose assembly and disassembly can be con-trolled in a stimulus-responsive manner[34–40], we have developed an in vitro strategy for producing lumazine synthase-derived nucleocapsids in response to an external cue. Fusion of an auxiliary protein to the

[1]Laboratory of Organic Chemistry, ETH Zürich, Zürich, Switzerland. [2]Present address: Institute of Biomaterials and Bioengineering, Tokyo Medical and Dental University, Chiyoda-ku, Tokyo, Japan. [3]Present address: École Polytechnique Fédérale de Lausanne (EPFL), SB ISIC LIBN, Lausanne, Switzerland. [4]Present address: MRC Laboratory of Molecular Biology, Francis Crick Avenue, Cambridge, UK. [5]These authors contributed equally: Mao Hori, Angela Steinauer. ✉e-mail: hilvert@org.chem.ethz.ch

cage subunit via a cleavable linker enables the isolation of otherwise aggregation-prone monomers in soluble form, sterically blocking both the formation of nonspecific aggregates and self-assembly into ordered capsid structures. Upon removal of the steric block by the addition of a sequence-specific protease, RNA-templated cage assembly can proceed. During in vivo assembly, the packaging of RNA requires specific packaging signals to outcompete interactions of host RNA with the capsid protein[33]. In vitro, these signals are less important. As a result, a broader range of RNA cargo molecules can be successfully packaged than is possible in vivo, underscoring the usefulness of this approach.

## Results

### Design and preparation of a sterically blocked NC-4 monomer

The nucleocapsid NC-4 efficiently packages its own encoding mRNA in vivo[33]. It was generated from the bacterial enzyme lumazine synthase by design and multiple rounds of directed evolution[32,33]. Each monomer is composed of a circularly permuted cage-forming subunit and an N-terminal RNA-binding peptide called λN+. Evolutionary optimization led to dramatic structural changes in subunit packing to yield an interlaced 240-subunit icosahedral capsid that is impermeable to nucleases. In addition, a robust RNA stem-loop packaging cassette that emerged in the mRNA sequence ensured high encapsidation yields and specificity in the competitive environment of the cellular cytosol.

Recapitulating NC-4 nucleocapsid formation in vitro would be attractive as a means to investigate both assembly and packaging mechanisms and for efforts to repurpose this system as a delivery vehicle for a wide range of RNA cargo molecules. However, our preliminary attempts to assemble these capsids from their component building blocks in the test tube failed because the isolated protein subunits tend to aggregate and form insoluble precipitates. We envisaged that fusion of an auxiliary protein to the NC-4 monomer via a cleavable linker might prevent these problems, solubilizing the protein and sterically blocking spontaneous assembly until triggered by an appropriate external stimulus (Fig. 1). A similar strategy has proved useful for the controlled assembly of bacterial microcompartment proteins into a range of distinct architectures[34].

As an auxiliary protein, we chose maltose-binding protein (MBP) because of its bulkiness and high solubility[41]. We genetically fused MBP to the N-terminal λN+ peptide of the NC-4 monomer via a linker containing a TEV protease cleavage site (Fig. 2a and Supplementary Table 1). We call the resulting construct MBP-NC-4. The fusion construct was produced in *Escherichia coli* and, following cell lysis in 50 mM phosphate buffer (pH 8.5) containing 1 M NaCl to minimize RNA binding to the cationic λN+ peptide, purified by affinity and size-exclusion chromatography (SEC) (Fig. 2b). Based on the SDS-PAGE analysis and SEC retention volume (Fig. 2c), purified MBP-NC-4 was identified as a soluble, monomeric protein with a molecular mass of

~66 kDa. Furthermore, the OD 260/280 ratio of 0.53 indicated the absence of contaminating nucleic acids. The purified protein was stable in solution for days, with no evidence of aggregation or precipitation.

To test the responsivity of the TEV cleavage site between MBP and NC-4, the fusion protein (48 μM) was incubated in 50 mM phosphate buffer (pH 7.4) containing 200 mM NaCl with 0.1 U/μL TEV protease and 1 mM DTT at room temperature overnight. SDS-PAGE analysis indicated complete cleavage of the fusion protein into two fragments corresponding to MBP (~44 kDa) and the NC-4 monomer (~21 kDa) (Fig. 2c). The identity of the latter was confirmed by nano-electrospray ionization (ESI) mass spectrometry (calculated average mass, 21,453.23 Da; found, 21,453.00 Da; Supplementary Fig. 1). In the absence of RNA cargo, TEV cleavage did not induce capsid assembly, only aggregation and precipitation of the NC-4 protein (Supplementary Fig. 2), consistent with the preliminary in vitro trials with the non-fused protein monomer. RNA is apparently needed to template and guide proper assembly.

### Stimulus-induced in vitro reconstitution of NC-4 nucleocapsids

With a stable fusion protein in hand, in vitro, the reconstitution of NC-4 capsids was investigated in the presence of RNA cargo. The native NC-4 mRNA containing the evolved stem-loop packaging cassette[33] was prepared by in vitro transcription. When mixed with 96 molar equivalents of the MBP-NC-4 fusion protein, it forms a complex, visible by native agarose gel electrophoresis (AGE) (Supplementary Fig. 3, lane 16), that is presumably mediated by electrostatic interactions between the polyanionic mRNA and the cationic λN+ peptide of NC-4. This particular stoichiometry, which was chosen to match the 96:1 monomer-to-RNA molar ratio observed for NC-4 capsids assembled in vivo, fully complexes all the RNA in the sample.

Nucleocapsid assembly was triggered by TEV protease cleavage of the fusion protein in the presence of RNA and monitored by native AGE. Since electrostatic interactions are the main driving force for RNA binding to the λN+ peptide, ionic strength should have a large impact on NC-4 assembly and was therefore optimized by varying the NaCl concentration in the assembly buffer. The MBP-NC-4 fusion protein and the in vitro transcribed RNA were mixed in 50 mM phosphate buffer containing 150, 300, 600, or 800 mM NaCl (pH 7.4). Capsid formation was induced after 1 h incubation at room temperature by adding 0.1 U/μL TEV protease and 1 mM DTT, and the resulting reaction mixtures were incubated overnight at room temperature.

NC-4 capsids assembled in vivo protect encapsulated RNA from nucleases[33]. To ensure that regular shell-like structures are also formed in vitro, the samples were challenged with either benzonase (2.5 U/μL) or RNase A (10 μg/mL) under the same conditions used in the original evolution experiments[32,33] (Supplementary Fig. 4). Samples containing high salt concentrations gave smeared bands of the approximately correct size for the nucleocapsid, which disappeared upon nuclease

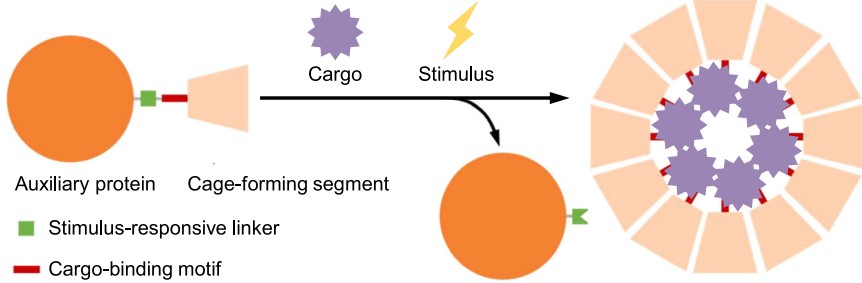

**Fig. 1 | Stimulus-responsive formation of cargo-containing protein cages.** The precursor protein consists of the cage subunit linked to an auxiliary protein via a stimulus-responsive linker and a cargo-binding motif. Cleavage of the auxiliary protein by an external stimulus enables self-assembly of the cage subunit and spontaneous cargo encapsulation.

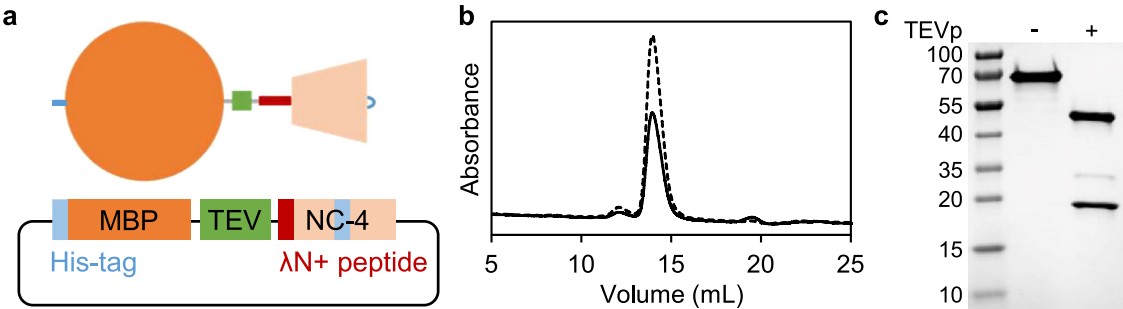

**Fig. 2 | Design and characterization of the sterically blocked NC-4 fusion protein. a** A His-tagged maltose binding protein (MBP, orange circle) was appended to the N-terminus of the NC-4 subunit via a linker containing a TEV protease cleavage site (TEV, green square). NC-4 includes the RNA-binding λN+ peptide (red rectangle), a cage-forming segment (beige trapezoid), and a His-tag in an external surface loop (blue line). The plasmid encoding the NC-4 fusion protein is shown schematically at the bottom of the panel. **b** SEC trace of the NC-4 fusion protein expressed in *E. coli* and purified by Ni-NTA column chromatography showing the absorbance trace at 260 nm (solid line) and 280 nm (dashed line). The NC-4 fusion protein elutes at 14 mL, consistent with the expected molecular mass (~66 kDa) of a monomer. **c** SDS-PAGE of the NC-4 fusion protein after FPLC purification. Before TEV protease (TEVp) treatment (−), only a thick band for the full-length NC-4 construct (~66 kDa) is seen. After TEV protease treatment (+), three bands corresponding to the MBP fragment (~44 kDa), TEV protease (~28 kDa), and the NC-4 monomer (~21 kDa) are observed, showing the responsivity of the NC-4 fusion protein to TEV protease. The experiment was independently repeated three times with similar results.

treatment (Supplementary Fig. 4, lane 10–15), likely because the high ionic strength disrupts the protein-RNA interactions needed to orchestrate and maintain proper capsid assembly. In contrast, well-defined bands on agarose gels close to that of in vivo assembled NC-4 were observed for the samples containing 150 and 300 mM NaCl, suggesting successful nucleocapsid assembly (Supplementary Fig. 4, lanes 4–9). While these assemblies were stable to benzonase, a 60-kDa dimeric nuclease, band intensity decreased substantially upon treatment with the smaller, 14-kDa RNase A. If incomplete assembly yielded any gaps in the shell wall, RNase A would be expected to access the lumenal space and degrade cargo RNA, leading to the disassembly of the protein cage.

With the aim of improving capsid yield, we explored whether increasing the amount of fusion protein relative to RNA could be used to drive more complete assembly. To that end, MBP-NC-4 and the NC-4 mRNA were mixed at the same initial molar ratio of 96:1 and, after 1 h pre-incubation, assembly was initiated as before by the addition of TEV protease and DTT. The salt concentration was set to 300 mM, which gave the best yields of RNase A-resistant nucleocapsids in the previous experiments (Supplementary Fig. 4, lane 9). After overnight incubation, an additional 13, 20, or 26 equivalents of the fusion protein relative to RNA were added, together with fresh TEV protease and DTT, to fill any remaining gaps in the capsid shells. After incubation at room temperature for another 24 h, the reaction mixtures were subjected to the nuclease challenge and analyzed by native AGE. As seen in Supplementary Fig. 5, the second addition of capsid subunits resulted in higher yields of nuclease-resistant nucleocapsids, consistent with the hypothesis that a substantial fraction of incomplete shells was still present after the first incubation period. Increasing the number of additional equivalents beyond 13 did not improve the yields, however.

### Biochemical and structural characterization of in vitro assembled NC-4

For biochemical characterization, the in vitro assembled NC-4 nucleocapsids were prepared with the optimized protocol described in the previous section, qualitatively assessed by native agarose gel electrophoresis (AGE) (Fig. 3a), and purified by anion-exchange chromatography (Fig. 3b). The fractions corresponding to the major peak (~43 mL) were collected to determine RNA packaging efficiency. The number of encapsulated nucleotides per cage was calculated by UV spectroscopy[42] (Fig. 3c).

NC-4 assembled in vivo encapsidates ca. 2500 nucleotides in total, corresponding to two to three copies of the 863-nt long capsid mRNA plus some *E. coli*-derived RNAs[33]. In contrast, the average number of nucleotides packaged in vitro is lower, ca. 1500 nt, which accounts for the nucleocapsid's slower migration on native agarose gels[43–51] (Fig. 3a). The lower total RNA content reflects the absence of *E. coli* RNA and hence a cleaner product for further use. Denaturing urea PAGE analysis of the extracted RNA confirmed that only the full-length target mRNA was encapsulated (Fig. 3d). Based on the length of the cargo sequence, most capsids must therefore contain two cargo molecules, with a smaller subpopulation harboring only one. Furthermore, by comparing the amount of RNA recovered in the particles with the original input RNA, we estimate that 26% of the sample was successfully packaged in vitro, surviving both nuclease treatment and ion exchange chromatography.

Transmission electron microscopy (TEM) images show that the in vitro assembled particles are uniform in size with a diameter of ~30 nm (Supplementary Fig. 6). Characterization of these particles by cryo-electron microscopy (cryo-EM) confirms that they adopt the same 240-subunit, icosahedrally symmetric, T = 4 structure as the NC-4 particles produced in vivo[33] (and Supplementary Fig. 7). In both cases, domain-swapped trimeric building blocks (Fig. 4) give rise to a tightly enlaced, closed-shell structure that effectively excludes nucleases from the lumenal space.

### Encapsulation of non-cognate RNA

The evolution of a packaging cassette in the mRNA encoding NC-4 was key to efficient and selective encapsidation in vivo[33]. In the absence of competing host nucleic acids in vitro, the specific set of packaging signals might be less important, which would facilitate binding a wider range of RNA guests. To investigate this possibility, we tested cargo molecules of different sequences and lengths for their ability to template capsid assembly upon TEV cleavage of the MBP-NC-4 fusion protein. The specific constructs were: (1) the 15 nt boxB RNA hairpin that binds the λN+ peptide with picomolar affinity[52,53]; (2) a 496 nt-long mRNA encoding HIV protease; (3) a 572 nt-long HIV protease mRNA equipped with 5′ and 3′ boxB tags, which had been used in the first round of nucleocapsid evolution[32]; and (4) a tandem 1175 nt-long mRNA encoding both the NC-4 capsid protein and HIV protease flanked by 5′ and 3′ boxB tags (Supplementary Table 2).

Packaging experiments were carried out according to the optimized protocol for NC-4 and monitored by native AGE, keeping the final concentration of the fusion protein and the total number of nucleotides constant. The results are summarized in Fig. 5 and Supplementary Fig. 8. Despite its high affinity for the λN+ peptide, the 15-nt

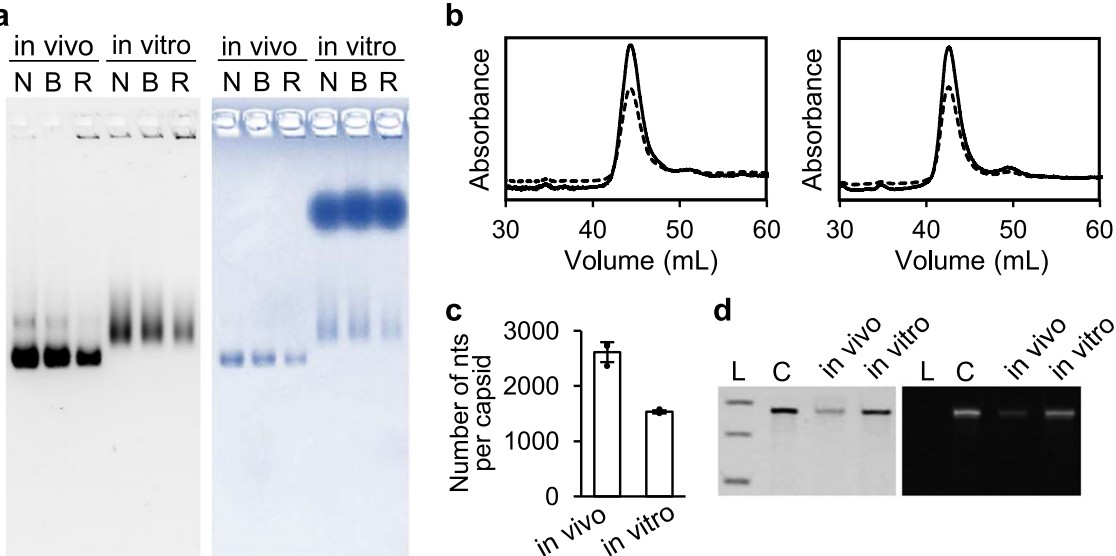

**Fig. 3 | Characterization of in vitro assembled NC-4 nucleocapsids. a** Native-AGE image of the in vivo and in vitro assembled NC-4 stained with GelRed (left) and Coomassie Brilliant Blue (right) (N: no treatment, B: benzonase treatment, R: RNase A treatment). The experiment was independently repeated three times with similar results. **b** Anion exchange chromatography traces of the in vivo (left) and in vitro (right) assembled NC-4 after RNase A treatment showing the absorbance at 260 nm (solid line) and 280 nm (dashed line). **c** The number of nucleotides (nts) per cage was calculated based on the UV absorbance[42] and presented as mean values +/− SD ($n = 3$, independent triplicate experiments). **d** Denaturing urea PAGE gel (6% 19:1 acrylamide/ bis-acrylamide, 8 M urea) images of the RNAs extracted from in vivo and in vitro assembled NC-4 after RNase A treatment and purification stained with GelRed (left), which stains all RNA, and the fluorogenic dye DFHBI-1T (right), which selectively binds the broccoli aptamer present in the 5′- and 3′-untranslated regions of the NC-4 mRNA (L: RNA ladder corresponding to 1000, 500, 300 nt; C: in vitro transcribed NC-4 mRNA as a control). The experiment was independently repeated three times with similar results.

boxB hairpin does not promote the formation of nucleocapsids on its own (Supplementary Fig. 8, lanes 4–6). The hairpin is too short to bind more than one coat protein, and simple charge neutralization of the cationic tag is apparently insufficient to nucleate proper assembly of the capsid subunits.

Consistent with the idea that effective RNA cargo must have a minimal length, an mRNA that encodes HIV protease (496 nt) but lacks both engineered and evolved packaging signals successfully yielded RNase A-resistant nucleocapsids (Fig. 5 and Supplementary Fig. 8, lane 9). The purified particles contained ~1600 nt, estimated from UV–Vis analysis[42], which corresponds to the encapsulation of three full-length mRNA molecules. Denaturing urea PAGE analysis of the RNA extracted from the capsids revealed only a single, sharp band with a MW of ~500 nt, in good agreement with the input mRNA. Appending flanking boxB packaging motifs to the HIV protease mRNA enhanced packaging efficiency somewhat. In this case, the total number of nucleotides increased to ~2300 nt, which corresponds to the encapsulation of four copies of the slightly longer 572-nt mRNA per particle (Fig. 5c and Supplementary Fig. 8, lane 12). In addition, the fraction of packaged mRNA improved from 6% for the 496 nt-long transcript to 16% for the 572 nt-long transcript, calculated based on total input RNA.

The most efficiently packaged cargo molecule was the long mRNA encoding both NC-4 and HIV protease in a single 1175-nt tandem transcript (Fig. 5a–c and Supplementary Fig. 8, lane 15). In addition to the evolved packaging cassette in the NC-4 gene, this construct has engineered boxB motifs at the 5′ and 3′ ends of the HIV protease gene. The resulting nuclease-resistant particles contained ~2600 nt, which corresponds to the encapsulation of approximately two full-length copies of an RNA that is ~30% longer than the original NC-4 mRNA. The overall encapsulation efficiency was 28%, which is comparable to the 26% efficiency observed for the evolved NC-4 nucleocapsids assembled in vitro.

The ability to encapsulate noncognate RNA greatly increases the utility of the evolved NC-4 capsid system. Negative stain TEM and cryo-EM show that the resulting particles have the same structure and

morphology as the NC-4 capsids assembled in vivo (Fig. 5b), even though the elution times in anion exchange chromatography depend on RNA cargo size (Fig. 5a). The latter phenomenon has been observed for adeno-associated virus particles and exploited to quantify and separate empty and full capsids[43–51]. Although explicit packaging instructions are not required for encapsidation in vitro, our results show that the presence of packaging signals and longer RNA lengths both have a positive effect on packaging efficiency. Dissection of the contributions of individual packaging signals and RNA length to efficiency will require more systematic investigation but optimizing such features through design[54] represents a tantalizing opportunity for further improving the assembly of nucleocapsids containing any desired RNA.

## Discussion

In this study, we have successfully recapitulated the assembly of nuclease-resistant NC-4 nucleocapsids in vitro. Although these nucleocapsids assemble efficiently in vivo and are well-behaved when isolated, the subunits on their own tend to aggregate non-specifically and precipitate in the absence of RNA, making their isolation and purification difficult. Attaching a large auxiliary protein to the N-terminus of the capsid subunit solves this problem by solubilizing the monomer and sterically blocking assembly, allowing efficient production and purification of the precursor protein. If the auxiliary protein is connected to the capsid-forming domain via an enzyme-cleavable linker, it can be easily removed by the addition of a sequence-specific protease, so that RNA-mediated assembly of the protein cage can be initiated on demand.

The high specificity and control over the assembly process afforded by this stimulus-response strategy facilitated optimization of in vitro assembly, mirroring the results seen for protease-triggered assembly of bacterial microcompartments[34]. The NC-4 system differs from the latter in that proteolytic removal of the bulky auxiliary protein produces the cage subunits as monomers as opposed to pre-formed, hexameric capsomers, providing a window on early events in

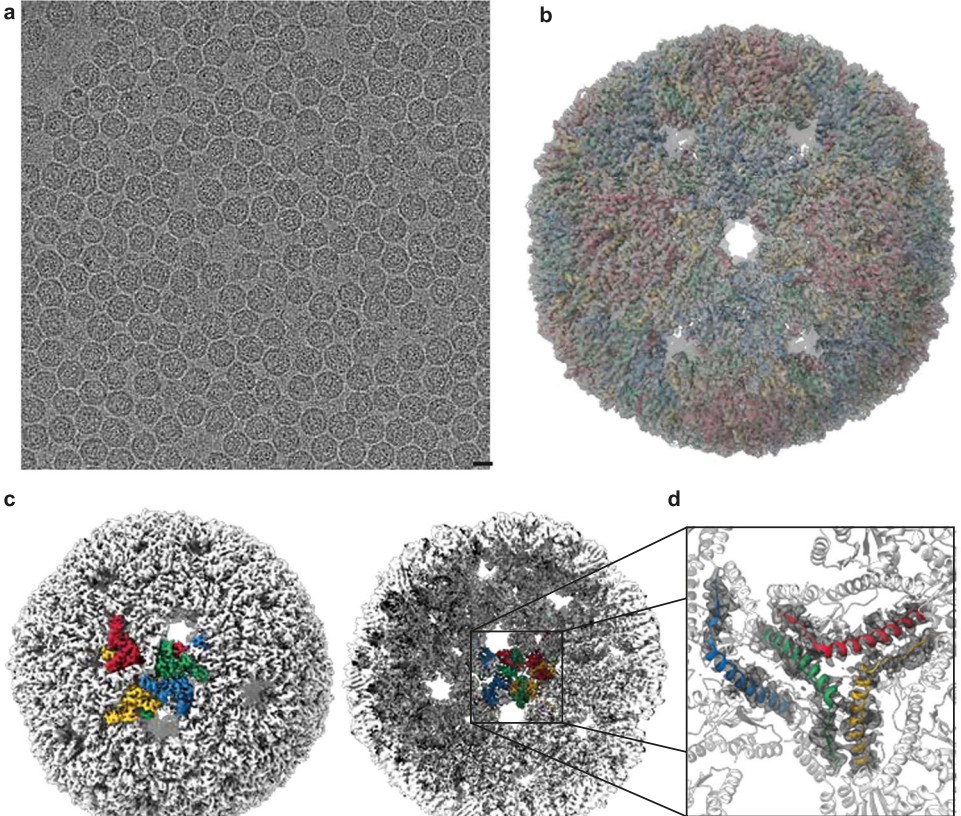

**Fig. 4 | Cryo-EM characterization of in vitro-assembled NC-4. a** Sample image of vitrified NC-4. Representative micrographs from 5019 collected movies, collected on a single biological replicate. Scale bar, 20 nm. **b** A fit of the previously published, in vivo-assembled capsid (PDB-ID 7a4j) into the obtained 3.5 Å map (gray mesh, contour level 0.021) shows that capsids assembled in vitro maintain the same structure as their counterparts produced in vivo. The four quasi-equivalent chains are colored red, yellow, green, and blue. **c** The map obtained from in vitro-assembled NC-4 cages (contour level 0.021) with four quasi-equivalent subunits colored as in (**b**), shown from the exterior (left) and the interior (right). **d** A view from the capsid interior, with only residues 55–95 highlighted in color. These residues form a hinge that enabled a domain swap between neighboring subunits during the laboratory evolution of NC-4. Restricting the cryo-EM map (dark gray mesh, contour level 0.0162) to the same area shows that the in vitro-assembled capsids feature the same domain-swapped structure as NC-4 isolated from cells.

nucleocapsid formation. Ionic strength proved to be critical for NC-4 assembly. Notably, pre-incubation of the fusion protein with the cargo RNA also improved assembly performance substantially. As depicted in Fig. 6, the RNA likely binds to the cationic λN+ peptide linking NC-4 and MBP, loosely pre-organizing the fusion proteins. After cleavage by TEV protease, detachment of the auxiliary protein allows the capsid subunits to assemble around the RNA. This step is critical, as no protein cages were formed in the absence of RNA or with the 15 nt-long boxB RNA hairpin, which binds the cationic λN+ peptide specifically and with high affinity but is too short to nucleate assembly. These observations underscore the crucial role played by RNA in the successful assembly of this system, where it serves as a template to recruit multiple monomers and orchestrate their productive interaction. Cooperative assembly of capsid proteins around viral RNA is similarly a hallmark of many natural viruses[55–63].

During the evolutionary optimization of NC-4, the simple boxB packaging tags that had been included at both ends of the encoding mRNA were supplanted by a more sophisticated cassette of RNA stem-loops, consisting of three boxB-like stem-loops with lower λN+ affinity clustered around the high-affinity 5′ boxB tag, which likely guided the specificity and efficient assembly of the capsids in the competitive environment of the bacterial cytosol[33]. In the absence of competing host RNA in vitro, RNA cargo lacking packaging signals can also be encapsidated. Although somewhat less efficient than assembly mediated by cargo possessing packaging signals, simple electrostatic interactions between the negatively charged RNA molecule and the positively charged RNA-binding motif on each capsid monomer

evidently suffice to induce the formation of properly formed nucleocapsids. Consequently, this system may serve as a versatile encapsulation tool for a potentially wide range of RNA molecules, including therapeutic RNAs, as well as other negatively charged macromolecules.

The inherent modularity of the fusion protein, consisting of a capsid-forming component, an RNA binding tag, and a steric block, opens up possibilities for tailoring the properties of this system for a range of applications. For example, the λN+ peptide could be replaced with a different binding motif to enable selective encapsulation of molecules other than RNA, including DNA, proteins, synthetic polymers, and inorganic complexes. The homogeneous size and morphology of AaLS-derived capsids provide an advantage over hollow nanoparticles composed of lipids or synthetic polymers, especially for biomedical applications, for which strict quality control is required[64]. Finally, any molecule that is large enough to prevent the cage-forming segment from assembling could be used as an auxiliary protein. Enzymes for prodrug therapy and antibody fragments targeting specific cells/tissues are possible candidates that would enable combination therapy and targeting.

In sum, stimulus-responsive assembly of nonviral nucleocapsids provides a powerful tool for both basic and applied research. As a versatile encapsulation platform, it offers exciting opportunities to probe assembly mechanisms and package diverse cargo molecules on demand. Extension to other viral and nonviral protein cages will further contribute to biological, chemical, and medical research by providing customizable vehicles for a wide range of delivery, vaccine, and other therapeutic applications.

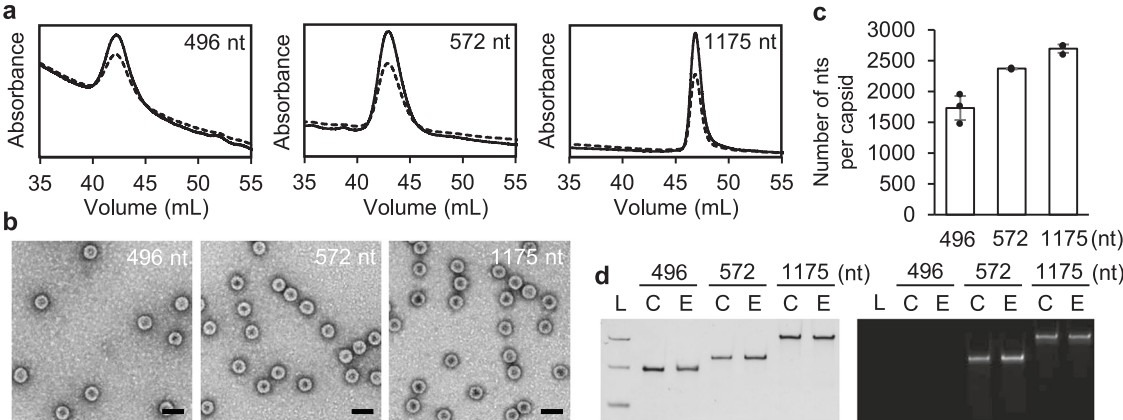

**Fig. 5 | Characterization of in vitro assembled NC-4 nucleocapsids containing different RNA cargo. a** Anion exchange chromatography of samples prepared with genes encoding HIV protease without boxB tags (496 nt, left), HIV protease equipped with boxB tags (572 nt, middle), and both NC-4 and HIV protease equipped with boxB tags (1175 nt, right) after RNase A treatment. Absorbance traces at 260 nm (solid line) and 280 nm (dashed line) are shown. **b** TEM images after FPLC purification (scale bar: 50 nm). The experiment was independently repeated three times with similar results. **c** The number of nucleotides (nt) per capsid was calculated based on the UV absorbance[42] and presented as mean values +/− SD (*n* = 3, independent triplicate experiments). **d** Denaturing urea PAGE gel (6% 19:1 acrylamide/bis-acrylamide, 8 M urea) images of the RNAs extracted from the in vitro assembled nucleocapsids after the RNase A treatment and the purification stained with GelRed (left) and DFHBI-1T (right) (L: RNA ladder corresponding to 1000, 500, 300 nt, C: in vitro transcribed RNA control, E: extracted RNA). The experiment was independently repeated three times with similar results.

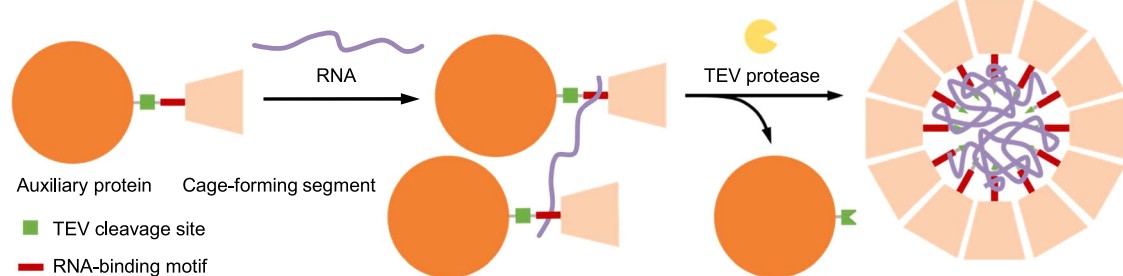

**Fig. 6 | Modified mechanism of in vitro assembly of NC-4 nucleocapsids.** RNA interacts with the RNA-binding motif of the NC-4 fusion protein that helps nucleation of the assembly of the cage-forming segment, although, at this stage, the auxiliary proteins still prevent cage formation (middle). After TEV protease treatment, the auxiliary proteins detach from the cage-forming segment and the protein cages assemble around cargo RNA.

## Methods

### Plasmid construction

To create the pQE_H6-MBP-NC4 plasmid, the NC-4 gene was amplified from the pMG_NC4 plasmid[33] and cloned into the pQE-H6-MBP vector[24] derived from pQE-80L (QIAGEN GmbH, Hilden, Germany). Briefly, a PCR product encoding NC-4 was amplified with primers containing flanking BamHI and HindIII restriction sites, 5′-GCG GCG GGA TCC GGA AAC GCG AGA ACG-3′ and 5′-GCG GCG AAG CTT GAG TTA GCC TCT GAT GAG-3′. The PCR-amplified fragment and the pQE-H6-MBP plasmid were digested with BamHI-HF and HindIII-HF (New England Biolabs, Ipswich, MA, USA) and ligated with T4 DNA ligase (Thermo Fisher Scientific, Waltham, MA, USA).

The plasmids used for the preparation of the non-cognate RNAs were prepared analogously from the plasmids pAC-Ptet-dBHIV[32] and pAC-Ptet-HIV-protease[20], which encode HIV protease with and without boxB tags, respectively. These plasmids were digested with XbaI and BlpI (New England Biolabs, Ipswich, MA, USA) and ligated into the pMG plasmid, which contains a T7 promoter and terminator[33], by using T4 DNA ligase. To construct the long mRNA encoding both NC-4 and HIV protease in tandem, a PCR-amplified fragment of HIV protease was prepared using pAC-Ptet-dBHIV[32] and primers containing flanking XhoI and BspEI restriction sites, 5′-GCG GCGCTC GAG CAT ATG CCT CAG ATC ACT C-3′ and 5′-GCG GCG TCC GGA CCC GGA TAT AG-3′. The PCR product and the pMG_NC4 plasmid were digested with XhoI and BspEI (New England Biolabs, Ipswich, MA, USA) and ligated using T4 DNA ligase.

The integrity of the open reading frame was confirmed by Sanger sequencing of the pQE_H6-MBP-NC4 plasmid with the standard primers QE-for (5′-GTA TCA CGA GGC CCT TT CG-3′) and QE-rev (5′-GTT CTG AGG TCA TTA CTG G-3′) and of the plasmids used for the preparation of the non-cognate RNAs with the standard primers T7 (5′-TAA TAC GAC TCA CTA TAG GG-3′) and T7-term (5′-TGC TAG TTA TTG CTC AGC GG-3′) (Microsynth, Balgach, Switzerland).

### Preparation and characterization of the MBP-NC-4 fusion protein

Chemically competent BL21(DE3) *E. coli* cells were transformed with the plasmid encoding the modified NC-4 monomer (Fig. 2a). In Lysogeny Broth (LB) (BD, Franklin Lakes, NJ, USA) containing 100 μg/mL ampicillin (PanReac AppliChem, Darmstadt, Germany), the cells were cultured at 37 °C until OD600 reached 0.6, then cooled to 25 °C and incubated overnight with 0.5 mM IPTG (PanReac AppliChem, Darmstadt, Germany). The cells were harvested by centrifugation at 15,000 × *g* and 4 °C for 15 min, and the resulting cell pellet was stored at −20 °C. For purification, the pellet from 200 mL culture was resuspended in 30 mL lysis buffer (50 mM sodium phosphate buffer containing 1 M NaCl and 20 mM imidazole, pH 8.5) and lysed by sonication on ice with a UP200S (Hielscher Ultrasonics GmbH, Teltow, Germany),

80% duty cycle and amplitude, and ON and OFF cycle of 1 min for 5 rounds.

After centrifugation at $8500 \times g$ and 4 °C for 15 min, the supernatant was loaded on 2 mL Ni-NTA agarose resin (QIAGEN GmbH, Hilden, Germany) in a gravity flow column. The column was washed with the lysis buffer followed by wash buffer (50 mM sodium phosphate buffer containing 1 M NaCl and 50 mM imidazole, pH 8.5), and bound proteins were eluted with elution buffer (50 mM sodium phosphate buffer containing 300 mM NaCl and 500 mM imidazole, pH 8.4). The pH of the eluted sample was adjusted to 7.4 and incubated at room temperature overnight. The sample was further purified by size-exclusion chromatography using a Superdex 200 increase 10/300 GL (GE Healthcare, Chicago, IL, USA) and storage buffer (50 mM sodium phosphate buffer containing 200 mM NaCl and 5 mM EDTA, pH 7.4; Fig. 2b). The fractions corresponding to the modified monomer were analyzed by SDS-PAGE (Fig. 2c). Briefly, the sample was incubated with NuPAGE LDS Sample Buffer (4X) (Thermo Fisher Scientific, Waltham, MA, USA) at 70 °C for 10 min. The resulting mixture and a protein molecular weight marker, PageRuler prestained protein ladder (Thermo Fisher Scientific, Waltham, MA, USA), were separated on a 4-20% Mini-PROTEAN TGX Precast Protein Gel (Bio-Rad Laboratories Inc., Hercules, CA, USA) at 200 V for 30 min using a Mini-PROTEAN Tetra Cell and PowerPac Universal Power Supply (Bio-Rad Laboratories Inc., Hercules, CA, USA). The gel was stained with a solution containing 0.1% Coomassie Brilliant Blue R-250, 40% MeOH, and 10% acetic acid with gentle agitation for 1 h. After destaining in 40% MeOH and 10% acetic acid, the gel image was recorded with a ChemiDoc MP Imaging System (Bio-Rad Laboratories Inc., Hercules, CA, USA). The SDS-PAGE analysis was also conducted for the modified NC-4 monomer treated with 0.1 U/μL of TEV protease in the presence of 1 mM DTT overnight at room temperature (Fig. 2c).

### RNA preparation

The 15 nt boxB RNA hairpin was purchased from Microsynth (Balgach, Switzerland). The other RNAs used in the assembly reactions were prepared by run-off transcription using a HighYield T7 RNA Synthesis Kit (Jena Bioscience, Jena, Germany). First, the plasmids encoding BoxBr-NC-4 monomer-BoxBr, HIV protease, BoxBr-HIV protease-BoxBr, and BoxBr-NC-4 monomer-HIV protease-BoxBr were used to prepare DNA templates as reported previously[33]. In vitro transcription reactions were set up following the manufacturer's instructions. After a 4 h incubation, DNA templates were removed using TURBO™ DNase (Thermo Fisher Scientific, Waltham, MA, USA), and the resulting RNAs were purified by LiCl precipitation. The RNAs encoding HIV protease with and without the BoxBr tags were further purified by preparative urea PAGE[33]. The purified RNAs were analyzed and quantified by urea-PAGE, a Qubit RNA HS Assay Kit, and NanoDrop 2000 (Thermo Fisher Scientific, Waltham, MA, USA).

### In vitro nucleocapsid assembly

The MBP-NC-4 (Supplementary Table 1) fusion protein and the NC-4 mRNA (Supplementary Table 2) were mixed at a molar ratio of 96:1 in 50 mM sodium phosphate buffer, pH 7.4, containing 300 mM NaCl (final concentration of the fusion protein: 48 μM). After pre-incubation for 1 h at room temperature, 1 mM DTT (Thermo Fisher Scientific, Waltham, MA, USA) and 0.1 U/μL TEV protease (New England Biolabs, Ipswich, MA, USA) were added and the reaction mixtures were further incubated overnight at room temperature. In total, 13 eq of the capsid monomer relative to the RNA cargo, 1 mM DTT, 0.1 U/μL TEV protease were added in this order and then incubated for 24 h at room temperature. The samples were treated with nucleases as previously described[33]. Briefly, 2.5 mM $MgCl_2$ and 2.5 U/μL benzonase (Merck KGaA, Darmstadt, Germany) or 10 μg/ mL RNase A (Merck KGaA, Darmstadt, Germany) were added and incubated at 37 °C for 1 h.

The samples were analyzed with native-AGE (2% agarose, TAE buffer, 110 V) and anion-exchange chromatography using a Mono Q 5/50 column (Cytiva, Marlborough, MA, USA), where the mobile phase consisted of the storage buffer containing 200–1000 mM NaCl[33]. The fractions corresponding to the nucleocapsids were collected and concentrated using Amicon Ultra-0.5 mL centrifugal filter units (MWCO 100 kDa). For the encapsidation of other mRNAs, the concentration of the fusion protein and the total number of nucleotides were maintained at the same levels as those used for the encapsidation of NC-4 mRNA.

### Analysis of the encapsulated RNA in the nucleocapsids

To determine the average number of RNA nucleotides per nucleocapsid, we first treated the samples with RNase A and purified them using anion-exchange chromatography[33]. This step ensured that the RNA detected was exclusively encapsulated within the nucleocapsids, having been protected from both RNase A degradation and anion-exchange chromatography. The concentrations of protein and RNA in these treated nucleocapsid samples were measured using a UV absorbance method commonly employed for analyzing the protein and RNA content in viruses and virus-like particles[42]. The efficiency of RNA encapsulation was then calculated by comparing the amount of RNA remaining post-treatment to the total RNA initially used in the in vitro assembly reaction. In addition, cryo-EM analysis confirmed that the in vitro assembled NC-4 nucleocapsids mirrored the structure of those assembled in vivo, consisting of 240 protein monomers. This allowed us to calculate the total number of nucleotides per nucleocapsid as the ratio of RNA concentration to protein concentration normalized to 240 monomers.

To assess the integrity of the encapsulated RNA, RNA was extracted from the nucleocapsids using an RNeasy Mini kit (QIAGEN GmbH, Hilden, Germany) as per the manufacturer's instructions. Consistently, analysis with a denaturing urea PAGE gel (6% 19:1 acrylamide/bis-acrylamide, 8 M urea) revealed a single, discrete band corresponding to the full-length target mRNA for each mRNA variant tested, with no detectable degradation observed (Figs. 3d and 5d). This purity of encapsulated RNA enabled the straightforward calculation of the number of mRNA molecules per capsid by dividing the total number of nucleotides by the length of the mRNA.

### Nano ESI-MS

SEC-purified MBP-NC-4 was treated with 0.1 U/μL TEV protease and then desalted using a C18 ZipTip (Millipore, USA) and analyzed in MeOH:2-PrOH:0.2% FA (30:20:50). The solution was infused through a fused silica capillary (ID75 μm) at a flow rate of 1 μLmin-1 and sprayed through ID30 μm Pico Tips (CoAnn Technologies, Richland WA, USA). Nano ESI-MS analysis of the samples was performed on a Synapt G2_Si mass spectrometer and the data were recorded with the MassLynx 4.2 Software (both Waters, UK). Mass spectra were acquired in the positive-ion mode by scanning an m/z range from 500 to 5000 Da with a scan duration of 1 s and an interscan delay of 0.1 s. The spray voltage was set to 3 kV, the cone voltage up to 50 V, and the source temperature to 100 °C. Recorded m/z data were deconvoluted into mass spectra by applying the maximum entropy algorithm MaxEnt1 (MaxLynx) with a resolution of the output mass 0.5 Da/channel and Uniform Gaussian Damage Model at the half height of 0.5 Da.

### Negative-stain transmission electron microscopy (TEM)

Carbon film-coated copper grids for TEM (Ted Pella, Inc., Redding, CA, USA) were negatively glow discharged at 25 mA for 45 s using the Emitech K100X (Quorum Technologies Ltd., Lewes, UK). In all, 8 μL of each sample was put on the grids individually. After 1 min incubation at room temperature, the excess solution was removed using filter paper and the grids were washed with double-distilled water (ddH$_2$O) twice.

The grids were dipped into 2% (w/v) uranyl acetate solution and excess solution was removed using filter paper immediately for the first time and after 10 s for the second time. The grids were air-died and observed with a JEM-1400 Plus (JEOL Ltd., Tokyo, Japan) operating at 120 kV.

## Cryo-electron microscopy

In vitro assembled NC-4 was analyzed by single-particle cryo-EM. Carbon-coated copper grids (R2/2, 400 mesh; Ted Pella, Inc.) were glow-discharged with an Edwards S150B sputter coater (BOC Edwards, Wilmington, MA, USA) for 30 s at 60% amplitude. A 3.5 μL droplet of the sample at a concentration of about 2 mg/mL protein was applied to the grid in a Vitrobot Mark IV (Thermo Fisher, Waltham, MA, USA) at 100 % humidity and 22 °C. It was then blotted at +25 blot strength for 13 s and plunge-frozen in liquid ethane.

The grids were analyzed on a 200 keV Glacios microscope (Thermo Fisher) equipped with a Falcon 3EC direct electron detector (Thermo Fisher), using EPU software (Thermo Fisher) for automated data collection. The data were processed in RELION 4.0 (ref. 65). Gain reference-corrected movies were motion and CTF corrected[66,67]. Particles were manually picked from a subset of images, and the best 2D classes from this subset of particles were used as input for autopicking over the entire dataset. An initial model was generated de novo by imposing icosahedral symmetry. 3D refinement, Bayesian polishing, and CTF refinement were performed to yield a 3.5 Å resolution structure.

## Reporting summary

Further information on research design is available in the Nature Portfolio Reporting Summary linked to this article.

## Data availability

The cryo-EM maps have been deposited in the Electron Microscopy Data Bank (EMDB) under accession code EMDB-16696. The raw cryo-EM movies have been deposited in the Electron Microscopy Public Image Archive (EMPIAR) under accession code EMPIAR-11973. Source data are provided in this paper.

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

## Acknowledgements

The authors are grateful to the Functional Genomics Center Zurich (FGCZ) and the ETH Scientific Center for Optical and Electron Microscopy for technical assistance and thank Mitsuru Naito and Nan Qiao (The University of Tokyo) for assistance in preparing SI Fig.4 and Thomas Edwardson for helpful discussions. This work was supported by ETH Zurich. M.H. was supported by an ETH Postdoctoral Fellowship 19-2 FEL-29 and A.S. received funding from the European Union's Horizon 2020 Research and Innovation Programme under Marie Skłodowska-Curie Grant Agreement 844006.

## Author contributions

M.H. and A.S. are responsible for the formulation of the research goals/aims, the stimulus-responsive self-assembly of protein cages, and in vitro assembly of nucleocapsids, respectively. M.H. collected all data except the ESI-MS spectrum, which was obtained by FGCZ, and the cryo-EM observation/analysis, which was performed by S.T. Preliminary experiments were conducted by A.S., J.H. and E.-M.M. under the supervision of A.S. with input by M.H. The manuscript was written by M.H. and D.H. and discussed and edited by the other authors.

## Funding

## Competing interests

The authors declare no competing interests.
