## [Peer Review File · Nature Communications]

REVIEWER COMMENTS

Reviewer #1 (Remarks to the Author):

The manuscript by Mao Hori et al. provides protease-triggered approach to prepare nonviral nucleocapsids. The study relies on a protease cleavable fusion protein consisting of maltose binding protein and NC-4 protein cage subunit, where the former acts as a steric block to prevent cage formation. Upon cleavage, the assembly and cargo loading of NC-4 are triggered. RNA cargo loading is studied in detail and digestion assays have been carried out to show that the cargo is protected to some extent. The merit of the study relies on the clever use steric block protein and the protease triggered assembly, which seems to work with high fidelity. However, the protection assay results are less impressive, and the actual benefit compared to existing approaches seems limited. Also, the release of the encapsulated cargo is not demonstrated. I find the study well-communicated, albeit sometimes difficult to follow due to missing links to figures. In this the text could be improved. The study is rather specific and in the absence of high cargo protection, release demonstration or other tailored properties outlined on lines 252-266, I find that the authors have not yet presented a compelling case to be published in Nature Communications.

Detailed comments:

1. Line 97 "identity of the latter was confirmed by nano-electrospray ionization (ESI) mass spectrometry". The mass spectrum should be shown in SI.
2. Line 100 "TEV cleavage did not induce capsid assembly, only aggregation and precipitation of the NC-4 protein (Supplementary Fig. 1)". These samples were filtered, but how about the others. If the NC-4 protein precipitates and is removed by filtration, how was the MS measurement possible?
3. Line 106: "When mixed with the MBP NC-4 fusion protein, it forms a complex, visible by native agarose gel electrophoresis (AGE)". How much of the fusion protein is needed for the complex formation? A constant amount of RNA should be titrated with the protein.
4. Line 115: "corresponds to the approximate RNA:protein charge ratio". It is not clear what the charge ratio is. Is it 1:1 or something else?
5. The in vitro assembled cages seem to provide a rather marginal protection for the encapsulated RNA against RNase. It would be useful to quantify band intensities and give a more quantitative analysis of the protection.
6. A question remains that why is there such a big difference in the protection of in vivo and in vitro assembled systems, especially if the resulting structures are supposed to be identical (as argued based on the cryo-EM data)?
7. Figure 3a is not cited or discussed in the text.
8. In Supplementary Fig. 4, why is the band intensity increasing upon benzonase treatment (Lanes 1 and 2, GelRed)?
9. Line 167: "In both cases, domain-swapped trimeric building blocks give rise to a tightly enlaced, closed-shell structure that effectively excludes nucleases from the luminal space". This does not seem to be true since the in vitro assembled structures are much more susceptible to nuclease degradation (see also my previous comment on this).
10. Figure 5a shows some important differences in peak positions depending on the cargo size (496 nt = 42 mL, 572 nt = 43 mL, 1175 nt = 47 mL). This data is not commented. Why is there such a difference? This would indicate differences between the particle size populations. Yet, the authors

claim that nsTEM data shows that all particles have the same structure and morphology.

11. Can the encapsulated RNA molecules be released from the capsids? Such demonstration would be important for the anticipated delivery applications.

12. Figure 6 is not cited or properly discussed in the text.

13. Lines 252-266. Here the authors present various interesting possibilities to widen the scope of the work. Any demonstration along these lines would increase the interest and impact of the work.

14. Figure captions do not describe sample size, error bars or provide statistical analysis details.

Reviewer #2 (Remarks to the Author):

This is a concisely written account of an interesting approach for controlling the in vitro formation of RNA-containing virus-like particles from evolved proteins that otherwise aggregate into insoluble precipitates. The idea is to solubilize the recombinant protein with a bulky expression tag – MBP in this instance – and then cut off the tag in the presence of RNA. The protein, including an N-terminal RNA-binding sequence (λ N+), is a lab-evolved version (NC-4) of the bacterial enzyme lumazine synthase, and in this work it is convincingly demonstrated that cutting off the MBP tag in the presence of RNA is necessary and sufficient for 240-subunit capsids to form in vitro around a range of different RNA molecules, including the NC-4-encoding mRNA itself. Systematic variation of salt concentrations and protein-to-RNA ratios lead to the identification of optimum conditions for nucleocapsid formation. The work is carefully designed and executed, and will stimulate further studies of this kind, in particular those involving capsid-forming/RNA-binding proteins that are not virus derived.

I recommend publication after the authors address the follow questions and concerns, none of them major, but all of them important nevertheless for their presentation to be clearer.

In lines 22-25, examples are given of molecular self-assembly underlying biological processes, e.g., the formation of lipid bilayers and biomolecular condensates. However “the folding of nucleic acid and protein polymers” are not examples of self-assembly, but rather of molecular conformational transitions – unless one is talking about the cooperative co-self-assembly of RNA and protein polymers, as in the present work and in viral nucleocapsids, in which case this should be made clear. Also, “macromolecular machines”-driven transcription and translation are active processes involving energy consumption and are as such not generally considered to be self-assembly examples.

Many times throughout the manuscript mRNA molecules are referred to as genomes, which they are not: see, for example, lines 57, 58, 64, 70, 104, 207, and 208. They are nothing more – or less – than mRNA molecules.

In lines 93-94 the absolute concentration of fusion protein incubated with 0.1 U/ul protease should be specified.

Similarly, in the line 134-147 paragraph, the effects on nucleocapsid-resistant yields of varying protein:RNA ratios are described; what about the effect of overall absolute concentrations?

In the line 154-162 paragraph it is suggested that the gel electrophoretic mobilities of nucleocapsids containing different amounts of RNA should be different. But why should they be different if the capsids have the same structure (and the same external charge)? For bromoviruses, for example, the mobility is independent of RNA content and is the same for empty capsids.

It would be helpful if line 199, referring to the change in packaging efficiency from 6% to 16%,

included a reminder to the reader that the two RNAs involved are the 496nt and 572nt RNAs mentioned several sentences earlier.

In the line 201-212 paragraph it is not easy to follow the discussion of how total-nt content and RNA-length content were determined – presumably, the former from UV-VIS analysis and the latter from extracted RNA gels with appropriate ladders. For example, in lines 204-205 it is written that “The resulting nuclease-resistant particles contained ~2600nt, which correspond to encapsidation of approximately two full-length copies of an RNA...” But is that full length 1175-nts, and if so where is that band in an extracted-RNA gel? And, what is the uncertainty in the determination of the ~2600nt?

In line 207 does the 26% packaging yield cited for the NC-4 mRNA refer to the in vivo or in vitro situation?

In line 263 the authors raise the intriguing possibility that a disease-specific protease could be used to induce cage assembly only at a disease-associated site: can they suggest an example? Further, can they suggest why it would be important to FORM RNA-containing nucleocapsids at these disease sites, rather than to DISASSEMBLE them?

Having established that the capsids can contain RNA contents as large as ~2600nt, why weren't experiments performed with ~2600nt-long RNA molecules?

In discussing the relative packaging efficiencies of different length shorter RNAs, like the 572-nt and 1175-nt RNAs, can the authors comment on the role of there being a different number of packaging signals involved in these molecules? Or on the role of different lengths on the capsid yields of the 496-nt and 572-nt RNAs (without and with packaging signals, respectively).

Finally, can the authors give more details of how much change there was in the RNA-binding and capsid-forming parts, respectively, as the lambdaN+/NC protein was evolved?

Reviewer #2:

The manuscript by Mao Hori et al. provides protease-triggered approach to prepare nonviral nucleocapsids. The study relies on a protease cleavable fusion protein consisting of maltose binding protein and NC-4 protein cage subunit, where the former acts as a steric block to prevent cage formation. Upon cleavage, the assembly and cargo loading of NC-4 are triggered. RNA cargo loading is studied in detail and digestion assays have been carried out to show that the cargo is protected to some extent. The merit of the study relies on the clever use steric block protein and the protease triggered assembly, which seems to work with high fidelity. However, the protection assay results are less impressive, and the actual benefit compared to existing approaches seems limited. Also, the release of the encapsulated cargo is not demonstrated. I find the study well-communicated, albeit sometimes difficult to follow due to missing links to figures. In this the text could be improved. The study is rather specific and in the absence of high cargo protection, release demonstration or other tailored properties outlined on lines 252-266, I find that the authors have not yet presented a compelling case to be published in Nature Communications.

We appreciate the reviewer's recognition of "the clever use steric block protein and the protease triggered assembly" as well as the constructive criticisms for clarifying the text. As outlined below, we believe that the concerns expressed about cargo protection are unwarranted as the in vitro and in vivo capsids are similar in this respect. We also disagree that our strategy has only limited benefit compared to existing approaches. While some model virus capsids can be disassembled into their component building blocks and reassembled under mild conditions, this is not the case for many other protein cages. Even a system as heavily researched as ferritin typically requires harsh pH changes for disassembly and assembly. The approach that we introduce solves this problem, yielding soluble, easily tractable monomers that can be assembled in a controlled manner, and can be readily extended to many supramolecular protein structures. Because our study provides novel insights into the assembly of artificial nucleocapsids and opens up potential applications in biotechnology and medicine, we believe that our manuscript warrants publication in Nature Communications.

Detailed comments:

1. Line 97 "identity of the latter was confirmed by nano-electrospray ionization (ESI) mass spectrometry". The mass spectrum should be shown in SI.

The requested mass spectrum has been added as Supplementary Fig. 1.

2. Line 100 "TEV cleavage did not induce capsid assembly, only aggregation and precipitation of the NC-4 protein (Supplementary Fig. 1)". These samples were filtered, but how about the others. If the NC-4 protein precipitates and is removed by filtration, how was the MS measurement possible

The MS sample was not filtered, but directly desalted using a C18 ZipTip (Millipore, USA) and analyzed in MeOH:2-PrOH:0.2% FA (30:20:50). These conditions are different than those used in our biochemical and FPLC analyses, which were carried out in phosphate buffer. The conditions for the different protocols are described in lines 93-103 and in the caption of Supplementary Fig. 1.

3. Line 106: “When mixed with the MBP NC-4 fusion protein, it forms a complex, visible by native agarose gel electrophoresis (AGE)”. How much of the fusion protein is needed for the complex formation? A constant amount of RNA should be titrated with the protein.

The goal of this study was to develop a new strategy for on-cue capsid assembly in vitro, rather than an in-depth biophysical characterization of the NC-4 system. The protein-to-RNA molar mixing ratio was chosen to match the ratio found in NC-4 nucleocapsids assembled in vivo. As shown in Supplementary Fig. 3 (lane 16), after the addition of the fusion protein, but before cleavage, free proteins but no free RNA transcripts are observed, suggesting that every RNA is fully complexed with protein under these conditions (lines 119-121). Although RNA titration might be informative in future detailed studies of our model system, we don't believe that it would improve encapsidation since RNA binding by the protein already appears to be saturated pre-cleavage.

4. Line 115: “corresponds to the approximate RNA:protein charge ratio”. It is not clear what the charge ratio is. Is it 1:1 or something else?

We appreciate the reviewer's bringing this point to our attention. In the original text, the ratio stated as 74:1 was a typographical error that did not correspond to our optimized conditions. The correct ratio is 96:1, which reflects the molar ratio of protein to RNA in the in vivo assembled capsid. The charge ratio is not relevant in this context. To clarify, we revised lines 113-121 and removed the reference to the charge ratio. We apologize for the confusion caused by the original formulation.

5. The in vitro assembled cages seem to provide a rather marginal protection for the encapsulated RNA against RNase. It would be useful to quantify band intensities and give a more quantitative analysis of the protection.

The in vitro-assembled cages do not provide “a rather marginal protection for the encapsulated RNA against RNase.” Both in vivo and in vitro-assembled cages show some RNA degradation in the presence of RNase but not benzonase, likely a reflection of the presence of some imperfectly assembled capsids. For particles that survive the RNase challenge, however, cargo protection appears to be excellent, as shown in Fig. 3d, and is similar to that observed for nucleocapsids that were assembled in vivo. The mRNA encapsulated in these particles is the same length as the in vitro transcribed RNA control and no fragments are observed on denaturing urea PAGE gels. The methods we used to calculate mRNAs per capsid are clarified in lines 165-178 of the main text and, in more detail, in the Methods section (lines 396-417).

6. A question remains that why is there such a big difference in the protection of in vivo and in vitro assembled systems, especially if the resulting structures are supposed to be identical (as argued based on the cryo-EM data)?

As explained in response to the previous point, there is no big difference in the protection of in vivo and in vitro assembled systems. Although incompletely assembled capsids are unstable to nuclease treatment, which influences the yield of correctly assembled capsids, once the particles are successfully assembled and isolated post RNase treatment, cargo protection is similar to that observed for capsids produced in vivo. See Fig. 3d and 5d, which show that only full-length RNA was extracted from purified NC-4 after RNase A treatment. Together with the total number of transcripts in each purified particle, these results

demonstrate that there is no significant difference in the protection afforded by the in vivo and in vitro assembled systems.

7. Figure 3a is not cited or discussed in the text.

Fig. 3a is now cited in lines 163 and 170.

8. In Supplementary Fig. 4, why is the band intensity increasing upon benzonase treatment (Lanes 1 and 2, GelRed)?

Prompted by the reviewer's comment, we revisited this experiment. When repeated, there was no increase observable in the band intensity. The differences observed in the previous experiment likely originated from a pipetting error. We consequently replaced the old Supplementary Fig. 4 with the new data.

9. Line 167: "In both cases, domain-swapped trimeric building blocks give rise to a tightly enlaced, closed-shell structure that effectively excludes nucleases from the luminal space". This does not seem to be true since the in vitro assembled structures are much more susceptible to nuclease degradation (see also my previous comment on this).

As explained in response to comments 5 and 6, the cargo quantification experiments show that the in vitro assembled structures obtained *following* RNase A treatment are not more susceptible to degradation than their counterparts assembled in vivo. The particles that degrade during the assembly process are presumably incomplete structures that still have large openings in the capsid shell.

10. Figure 5a shows some important differences in peak positions depending on the cargo size (496 nt = 42 mL, 572 nt = 43 mL, 1175 nt = 47 mL). This data is not commented. Why is there such a difference? This would indicate differences between the particle size populations. Yet, the authors claim that nsTEM data shows that all particles have the same structure and morphology.

The reviewer is correct. The peak positions of the capsids eluting from the ion exchange column vary with cargo size (Fig. 5a) even though the TEM data show that they have the same structure and morphology. We see no indication of differences between the particle size populations. Although perhaps surprising, this phenomenon has been well documented for adeno-associated virus capsids (doi: 10.3390/ijms232012332; 10.1016/j.omtm.2021.03.016; 10.1002/bit.28453; 10.1016/j.omtm.2021.04.003; 10.1016/j.omtm.2019.09.006; 10.1089/hgtb.2019.088; 10.1002/biot.202000015; 10.1089/hgtb.2011.217; <https://www.cytivalifesciences.com/en/us/solutions/cell-therapy/knowledge-center/resources/enhanced-aav-downstream-processing>). These studies show that the nucleic acid cargo can influence the elution time of viral capsids in anion exchange chromatography. Presumably, the difference in cargo size results in subtle variations in surface charge, or more precisely the zeta-potential, of the nucleocapsids, which in turn impacts the interaction of the particles with the ion-exchange matrix. We now comment on this observation at the end of the Results section (lines 226-237). The references are also cited in the discussion of the slower migration of in vitro-assembled NC-4 capsids on native agarose gels (line 170).

11. Can the encapsulated RNA molecules be released from the capsids? Such demonstration would be important for the anticipated delivery applications.

In this study, we released the encapsulated RNA molecules from the capsids by using a commercially available RNA extraction kit, RNeasy Mini Kit (Qiagen). See, for example, Figs. 3d and 5d. For many delivery applications, however, milder strategies for releasing the cargo inside cells will be needed. Various strategies for uncoating the protein shells are conceivable, including those employed by natural viruses (redox, metal ion or pH-triggered disassembly, selective proteolysis, etc.), but implementing them in the NC-4 system is beyond the scope of the current study. Here we establish a new strategy for assembling the NC-4 nucleocapsids in a stimulus-responsive fashion, which solves a practical problem associated with the tendency of the capsid subunits to aggregate and precipitate, thus opening opportunities to explore the NC4 assembly mechanism and the role of RNA packaging signals in the assembly process. We believe that this approach is significant in its own right as a potentially general strategy for controlling the formation of many protein assemblies.

12. Figure 6 is not cited or properly discussed in the text.

Fig. 6 was cited and discussed in lines 232-233 of the original manuscript and now in lines 258-266.

13. Lines 252-266. Here the authors present various interesting possibilities to widen the scope of the work. Any demonstration along these lines would increase the interest and impact of the work.

We believe that these experiments are beyond the scope of the current study, which 1) establishes a novel strategy to recapitulate the assembly of NC-4 nucleocapsids in vitro, 2) demonstrates the feasibility of encapsulating other cargo mRNAs, and 3) provides insight into the role of the evolved RNA packaging cassette.

14. Figure captions do not describe sample size, error bars or provide statistical analysis details.

The requested information has been added to the legends of Figs. 3c and 5c.

Reviewer #3:

This is a concisely written account of an interesting approach for controlling the in vitro formation of RNA-containing virus-like particles from evolved proteins that otherwise aggregate into insoluble precipitates. The idea is to solubilize the recombinant protein with a bulky expression tag – MBP in this instance – and then cut off the tag in the presence of RNA. The protein, including an N-terminal RNA-binding sequence (λ N⁺), is a lab-evolved version (NC-4) of the bacterial enzyme lumazine synthase, and in this work it is convincingly demonstrated that cutting off the MBP tag in the presence of RNA is necessary and sufficient for 240-subunit capsids to form in vitro around a range of different RNA molecules, including the NC-4-encoding mRNA itself. Systematic variation of salt concentrations and protein-to-RNA ratios lead to the identification of optimum conditions for nucleocapsid formation. The work is carefully designed and executed, and will stimulate further studies of this kind, in particular those involving capsid-forming/RNA-binding proteins that are not virus derived.

I recommend publication after the authors address the follow questions and concerns, none of them major, but all of them important nevertheless for their presentation to be clearer.

We appreciate the reviewer's thoughtful summary and support.

In lines 22-25, examples are given of molecular self-assembly underlying biological processes, e.g., the formation of lipid bilayers and biomolecular condensates. However “the folding of nucleic acid and protein polymers” are not examples of self-assembly, but rather of molecular conformational transitions – unless one is talking about the cooperative co-self-assembly of RNA and protein polymers, as in the present work and in viral nucleocapsids, in which case this should be made clear. Also, “macromolecular machines”-driven transcription and translation are active processes involving energy consumption and are as such not generally considered to be self-assembly examples.

We agree and have modified the introduction accordingly.

Many times throughout the manuscript mRNA molecules are referred to as genomes, which they are not: see, for example, lines 57, 58, 64, 70, 104, 207, and 208. They are nothing more – or less – than mRNA molecules.

We changed the text as requested.

In lines 93-94 the absolute concentration of fusion protein incubated with 0.1 U/ul protease should be specified.

The absolute concentration of the fusion protein (48 μ M) has been added to line 101.

Similarly, in the line 134-147 paragraph, the effects on nucleocapsid-resistant yields of varying protein:RNA ratios are described; what about the effect of overall absolute concentrations?

We did not systematically investigate the effect of overall absolute concentration. Although higher concentrations of the components might be expected to promote assembly, increasing the protein concentration when the molar ratios were varied (1:96 to 122) did not result in higher yields. Instead, as

shown in Supplementary Fig. 3 (Lane 13), higher protein concentrations promoted the formation of insoluble aggregates. This observation aligns with a recent study (<https://arxiv.org/abs/2307.04171>) showing that increasing the relative coat protein concentration in the assembly of RNA viruses did not increase the yield of icosahedral particles but led instead to the accumulation of malformed structures. These findings suggest that higher absolute concentrations may lead to undesirable outcomes rather than improved assembly efficiency and quality.

In the line 154-162 paragraph it is suggested that the gel electrophoretic mobilities of nucleocapsids containing different amounts of RNA should be different. But why should they be different if the capsids have the same structure (and the same external charge)? For bromoviruses, for example, the mobility is independent of RNA content and is the same for empty capsids.

The sensitivity of the gel electrophoretic mobilities of nucleocapsids containing different amounts of RNA mirrors the different elution profiles of the particles seen in Fig. 5. Although the mobility of bromoviruses may be independent of RNA capsids, as noted in our response to a similar query from Reviewer 1, studies on adeno-associated viruses have demonstrated that capsids with varying amounts of RNA (i.e., empty, partially loaded, and fully loaded capsids) exhibit differential migration during chromatographic separation methods, even though the capsids have similar structural morphology. As noted above in our response to Reviewer 1 (point 10), nucleic acid cargo has been shown to modulate the zeta potential, or the surface charge, of capsids, which could, in turn, affect their interaction with the gel matrix and therefore influence their migration during electrophoresis. Given the absence of an equivalent trend in bromoviruses, it is possible that this phenomenon might not be universal but rather specific to certain viral or artificial systems.

It would be helpful if line 199, referring to the change in packaging efficiency from 6% to 16%, included a reminder to the reader that the two RNAs involved are the 496nt and 572nt RNAs mentioned several sentences earlier.

This information has been added to lines 216-217.

In the line 201-212 paragraph it is not easy to follow the discussion of how total-nt content and RNA-length content were determined – presumably, the former from UV-VIS analysis and the latter from extracted RNA gels with appropriate ladders. For example, in lines 204-205 it is written that “The resulting nuclease-resistant particles contained ~2600nt, which correspond to encapsidation of approximately two full-length copies of an RNA...” But is that full length 1175-nts, and if so where is that band in an extracted-RNA gel? And, what is the uncertainty in the determination of the ~2600nt?

The total-nt content and RNA-length were determined by UV-VIS analysis and PAGE as described for the NC-4 capsids (lines 161-178) and in the Methods section (lines 396-417). For in vitro-assembled capsids isolated after RNase A treatment, we only observed single bands for the extracted RNAs in PAGE gels (Fig. 3d and 5d) which corresponded in size to the respective in vitro-transcribed control RNA. The total number of nucleotides encapsulated in the particles was determined by UV-Vis analysis using the method described by Zlotnick et al. (reference 43). The uncertainty in the determination is approximately 10% (error bars have been added to Figs. 3c and 5c) and is limited by the precision of the UV-Vis analysis.

In line 207 does the 26% packaging yield cited for the NC-4 mRNA refer to the in vivo or in vitro situation?

The reported packaging yield refers to the in vitro situation and the text was clarified accordingly.

In line 263 the authors raise the intriguing possibility that a disease-specific protease could be used to induce cage assembly only at a disease-associated site: can they suggest an example? Further, can they suggest why it would be important to FORM RNA-containing nucleocapsids at these disease sites, rather than to DISASSEMBLE them?

We've removed the original passage referencing induced cage assembly at a disease site to minimize speculation.

Having established that the capsids can contain RNA contents as large as ~2600nt, why weren't experiments performed with ~2600nt-long RNA molecules?

In vivo assembled capsids accommodate ~2500 nts (doi: 10.1126/science.abg2822). While experiments with longer transcripts could be performed, the RNA molecules that we chose for the current study were readily available in the lab. We felt that encapsulation of two copies of a 1175 nt-long RNA provided sufficient evidence for the loading capacity of these capsids.

In discussing the relative packaging efficiencies of different length shorter RNAs, like the 572-nt and 1175-nt RNAs, can the authors comment on the role of there being a different number of packaging signals involved in these molecules? Or on the role of different lengths on the capsid yields of the 496-nt and 572-nt RNAs (without and with packaging signals, respectively).

Judging from the fraction of RNA that is packaged and the number of mRNAs in the assembled particles. adding packaging signals and increasing cargo length both enhance packaging efficiency. However, these are qualitative trends, and disentangling their relative contributions will require more systematic study. Moreover, other factors, such as RNA sequence and secondary structure may also contribute to these outcomes. These points are now mentioned at the end of the Results section (lines 231-234).

Finally, can the authors give more details of how much change there was in the RNA-binding and capsid-forming parts, respectively, as the lambdaN+/NC protein was evolved?

As described in our recent paper (doi: 10.1126/science.abg2822), most of the mutations that appeared during evolution (14/17) were in the capsid-forming units or in the exterior loop introduced by circular permutation (2/17). Only one mutation occurred in the RNA-binding lambda N+ peptide, namely a Lys-to-Arg mutation that was known to increase the affinity of the RNA-peptide approximately threefold. However, as previously reported, reversion of this mutation had little effect on in vivo packaging.

REVIEWERS' COMMENTS

Reviewer #1 (Remarks to the Author):

The authors have addressed all my comments.

Reviewer #2 (Remarks to the Author):

I am satisfied with the responses from the authors to my questions and concerns from the original review.

I liked the paper then, and recommended publication then, and I like it still more now.